# Patient Experience in Home Respiratory Therapies: Where We Are and Where to Go

**DOI:** 10.3390/jcm8040555

**Published:** 2019-04-24

**Authors:** Cátia Caneiras, Cristina Jácome, Sagrario Mayoralas-Alises, José Ramon Calvo, João Almeida Fonseca, Joan Escarrabill, João Carlos Winck

**Affiliations:** 1Institute of Environmental Health (ISAMB), Faculty of Medicine, Universidade de Lisboa, 1649-028 Lisboa, Portugal; 2Healthcare Department, Praxair Portugal Gases, 2601-906 Lisboa, Portugal; 3CINTESIS-Center for Health Technologies and Information Systems Research, Faculty of Medicine, University of Porto, 4200-450 Porto, Portugal; cjacome@med.up.pt (C.J.); fonseca.ja@gmail.com (J.A.F.); 4Respiratory Research and Rehabilitation Laboratory (Lab3R), School of Health Sciences (ESSUA), University of Aveiro, 3810-193 Aveiro, Portugal; 5Service of Pneumology, Hospital Universitario Moncloa, 28008 Madrid, Spain; sarimayoralas@gmail.com; 6Healthcare Department, Praxair Spain, 28020 Madrid, Spain; jose_ramon_calvo@praxair.com; 7MEDCIDS-Department of Community Medicine, Health Information and Decision, Faculty of Medicine, University of Porto, 4200-450 Porto, Portugal; 8Allergy Unit, Instituto and Hospital CUF, 4460-188 Porto, Portugal; 9Hospital Clínic de Barcelona, 08036 Barcelona, Spain; ESCARRABILL@clinic.cat; 10Master Plan for Respiratory Diseases (Ministry of Health) & Observatory of Home Respiratory Therapies (FORES), 08028 Barcelona, Spain; 11REDISSEC Health Services Research on Chronic Patients Network, Instituto de Salud Carlos III, 28029 Madrid, Spain; 12Faculty of Medicine, University of Porto, 4200-319 Porto, Portugal; jcwinck@mail.telepac.pt

**Keywords:** Long-term oxygen therapy, home mechanical ventilation, patient-reported experience measures, quality of care, healthcare, sustainability

## Abstract

The increasing number of patients receiving home respiratory therapy (HRT) is imposing a major impact on routine clinical care and healthcare system sustainability. The current challenge is to continue to guarantee access to HRT while maintaining the quality of care. The patient experience is a cornerstone of high-quality healthcare and an emergent area of clinical research. This review approaches the assessment of the patient experience in the context of HRT while highlighting the European contribution to this body of knowledge. This review demonstrates that research in this area is still limited, with no example of a prescription model that incorporates the patient experience as an outcome and no specific patient-reported experience measures (PREMs) available. This work also shows that Europe is leading the research on HRT provision. The development of a specific PREM and the integration of PREMs into the assessment of prescription models should be clinical research priorities in the next several years.

## 1. Introduction

Long-term oxygen therapy (LTOT) and/or home mechanical ventilation (HMV) are well-established therapies for patients with chronic respiratory failure, such as those with chronic obstructive pulmonary disease (COPD), neuromuscular diseases, and obstructive sleep apnea (OSA), among others. These therapies represent key services in the home respiratory therapy (HRT) provided to these patients. Increasing numbers of patients receiving HRT are reported not only in Europe but also worldwide [1,2,3,4,5]. Thus, HRT is imposing a major impact on clinical care and healthcare systems. Over the next several years, the main challenge will be to ensure a sustainable healthcare system to continue to guarantee access to HRT while maintaining the quality of care.

According to the World Health Organization, quality of care is defined as “the extent to which health care services provided to individuals and patient populations improve desired health outcomes. In order to achieve this, health care must be safe, effective, timely, efficient, equitable and people-centered” [6]. A necessary step in the process of maintaining and improving quality is to monitor and evaluate the quality of healthcare in routine clinical practice. Based on the reactive, disease-focused, and biomedical model, the indicators of quality have been mainly restricted to traditional clinical metrics. A number of studies conducted over the last few decades have addressed the beneficial effects of HRT on morbidity, mortality, and adverse outcomes, as well as the variations in HRT provision among countries [5,7,8]. However, these metrics alone do not provide a complete picture of HRT quality.

The patient’s experience of treatment is a cornerstone of high-quality healthcare [9]. Only by analyzing the relational and functional aspects of the patient experience is it possible to assess the extent to which patients are receiving care that is in line with their preferences, needs, and values. The integration of the patient experience with healthcare delivery and quality evaluation are key steps in moving toward patient-centered and personalized care [10]. As Doyle et al. suggested, the patient’s experience is the third pillar of quality, along with clinical safety and effectiveness [11]. However, it is only in recent years that patients’ perceptions of healthcare provision have started to receive attention.

This review approaches the assessment of the patient experience in the clinical context of HRT while highlighting the European contribution to this emerging body of knowledge.

## 2. Patient Experience in the Context of HRT

The patient experience in the context of HRT is reviewed with a focus on two main areas: (1) HRT prescription models and the inclusion of the patient experience as an outcome of these models and (2) methods used to assess the patient experience. To address these two aims, a narrative review was conducted. The search, although not systematic in nature, included searches in electronic databases (PubMed, Medline, ISI Web of Knowledge and Google Scholar), as well as hand searches (expert consultation and a review of the reference lists in the included papers). The databases were searched between July and December 2018 using topic-related terms, such as oxygen therapy, home mechanical ventilation, noninvasive mechanical ventilation, home respiratory therapy, home treatment, chronic respiratory insufficiency, chronic respiratory failure, epidemiology, prescription, quality control, outcomes, patient experience, patient perspective, carers, caregivers, patient-reported experience measure, questionnaires, interviews, and focus groups. There was no time restriction in the literature search, although it was limited to English, Portuguese, or Spanish.

### 2.1. Prescription Models of HRT

There are a number of studies that have assessed the prescription of HRT. Table 1 summarizes 15 relevant studies on this topic. The majority of the studies (*n* = 9) were conducted from 2009 onward and primarily assessed the prescription of HMV (*n* = 10) [4,5,12,13,14,15,16,17,18,19], followed by LTOT (*n* = 6) [19,20,21,22,23,24]. The estimated prevalence of HMV (from 2.5 to 23/100,000 population) and of LTOT (from 31.6 to 102/100,000 population) were variable among distinct regions or countries. The estimated prevalence of HMV in Europe was 6.6 per 100,000 people, and Portugal was one of the countries with the highest prevalence [5].

Three studies reported the assessment of HRT prescription at a regional level (Catalan, Spain; Hong Kong, China; Tasmania, Australia), eight at a national level (Sweden, Canada, Poland, Denmark, England, Australia, France, Spain), and four at an international level (two countries, seven countries, 13 European countries, 16 European countries).

Most studies included both children and adult patients in their analysis. Only one of the studies specifically focused on a pediatric population [18]. Questionnaires, having been used in 10 studies, were the preferred method of data collection. In five studies, existing databases from HRT registries or health services were used. Irrespective of the data collection method used, data on users (age, sex, and diagnosis), type and duration of respiratory therapy, and equipment and interfaces were the most commonly recorded. None of the 15 studies reported the patient’s experience with HRT.

### 2.2. Assessment of Patient Experience

Assessing the patient experience has become a common approach to describing healthcare from the patient’s point of view, evaluating the process of care, and measuring the outcome of care [25,26,27]. Both quantitative and qualitative methods are being used to assess patients’ perception. Self-reported questionnaires, individual interviews, and focus groups are among the most frequently used methods of collecting data.

#### 2.2.1. Patient-Reported Experience Measures

The development of self-reported questionnaires, namely, patient-reported experience measures (PREMs) and patient-reported outcome measures (PROMs), has exponentially increased in the last several years. These two types of questionnaires collect information about the patient’s perspective but with distinct purposes. A PREM evaluates patients’ perception of their personal experience of the healthcare received, while a PROM assesses the perception of their health status and health-related quality of life [10,28]. A combination of PROMs and PREMs is essential to fully understand the performance of healthcare systems. Moreover, both measures are useful to provide a patient-centered perspective of healthcare, but PREMs are more adequate to assess experience with healthcare.

Distinct instruments to assess the patient’s experience with healthcare are available. Table 2 summarizes 14 instruments designed to assess the patient’s experience with the provision of care in different clinical settings [29,30,31,32,33,34], hospital [35,36,37,38], primary care [39,40], intermediate care [41], and community [33,41]. The majority of such instruments are generic and designed to be used for a diverse range of health conditions. However, two of the described questionnaires were specifically developed for patients with chronic diseases [29,34], and one was intended particularly for patients with COPD [30]. The majority of PREMs were developed to target adult patients and tested in patients who were at least 15 years old. Only two developed instruments were tested with the carers of children [31,39]. English is the most common language used, with some instruments also in Norwegian [31,38,39], Italian [35,41], and Spanish [29]. Most instruments already had some of their psychometric properties explored, namely, their reliability and validity.

None of the instruments above were specifically designed to assess the patient’s experience with HRT. However, a recent European Respiratory Society (ERS)/European Lung Foundation (ELF) survey was conducted across 11 European countries and assessed the attitudes and preferences of 687 patients on HMV and those of 100 carers [42]. A questionnaire was specifically developed for this study in eight languages (English, German, Dutch, Spanish, Italian, Portuguese, Greek, and French) and explored four areas: (1) patients’ demographic and clinical characteristics; (2) issues influencing compliance, such as interface comfort, abilities to travel, sleep, and socialize with a ventilator, type and technical functioning of the ventilator (e.g., alarms, ability to operate and change settings, on/off switches, and electricity consumption); (3) support, training, and education; and (4) requests for improved devices and support.

Today, it is possible to evaluate a patient’s perception of the HRT received using one of the described PREMs. Nevertheless, in the near future, the aim should be to develop a specific PREM to assess patients’ personal experience with HRT.

#### 2.2.2. Individual Interviews and Focus Groups

Qualitative studies that explore the experience of patients receiving HRT are still limited in the literature. Nevertheless, the literature review revealed some studies that explored the experience of patients living with COPD, pulmonary fibrosis, and OSA. These studies specifically focused on patients’ needs and the adaptation process to respiratory therapies. Two studies explored the patient’s experience with LTOT [45,46], and the others assessed the patient’s experience with non-invasive ventilation [47,48,49,50,51]. These studies were conducted in the United States of America [45,47], New Zealand [48,49], the United Kingdom [50], Sweden [51], and Spain [46] and included both adult patients and carers. Two reviews were also found on the needs of patients with COPD and were also used in the present analysis [52,53].

From the analysis of these studies, it was possible to clearly identify education, training, support, and carer involvement as important key-points in facilitating a patient’s treatment experience and subsequent adherence. Below, each one of these four key-points is described in detail.

Education: on the basis of the perspectives of patients, it is apparent that education is crucial for defining clear expectations about the treatment and motivating patient adherence. The main education topics raised by patients receiving respiratory therapies are related to disease self-management (e.g., COPD, OSA); physical effects and potential clinical benefits of the respiratory therapy; risks of not using the respiratory therapy; guidance on the use and function of equipment (e.g., continuous positive airway pressure (CPAP) devices, oxygen concentrators, how to use pulse oximeters and adjust flow with exertion); side effects and guidance on its management (skin protection, dry mouth, nasal congestion, irritated eyes); traveling with equipment; follow-up appointments; and assistance with financial elements (e.g., how to claim electricity costs) [45,46,49,50].

Training: formal training on appropriate equipment use has been suggested to be an important strategy for improving adherence [46,47,48,49,50,51]. Healthcare professionals need to introduce the device, explore possible practical problems, and give advice/help to solve these problems. In their initial experiences with respiratory therapy, patients should have a hands-on demonstration for setting up the device, trialing different masks/pressures, making mask adjustments, conquering different side-effects, and finding the best position for the tubing or machine (also considering the loudness of the device). Regular follow up visits or phone calls are important to assess practical problems being experienced (e.g., pressure from the mask, mask leakage, disturbing noise, and difficulties changing sleeping positions) and to discuss effective strategies to address them.

Support: establishing a trustworthy relationship with healthcare professionals after the initiation of respiratory therapy is perceived as helpful by patients, and these relationships positively influence their adherence [46]. Healthcare professionals need to foster a non-judgmental environment in which patients have opportunities to ask questions, share concerns and feelings, feel listened to, and feel understood. This is particularly important following the initiation of therapy [47], as questions or concerns are more likely to arise during the first days or weeks of treatment [49,52]. These opportunities can arise during regular follow-up visits, scheduled follow-up phone calls, and through access to a 24-h hotline [47].

Carer involvement: carers provide substantial care (emotional, physical) to the individual on a daily basis and, most of the time, live in the same house as the patient. On the basis of their important role in patients’ lives, carer involvement has been found to be essential to patients receiving HRT [45,46,47,48,50,51,52,53]. Patients recognize that carers play a major role in their treatment by helping them manage the disease and adapt to the equipment (e.g., verbal reminders, encouragement, setting up the machine, making mask adjustments, reassurance of therapy benefits). Carers themselves recognize their need for information regarding aspects of the disease and benefits of the HRT [47]. Carer involvement is thus perceived by all stakeholders as an essential component of education and training from the beginning of treatment [45,47,48,50,51,52,53], and it is generally associated with positive results, namely, the patients’ adoption and adherence to HRT [47,53].

## 3. Discussion

This comprehensive review is a first critical step toward the assessment of the patient experience in the clinical context of HRT. It demonstrates that research in this area is still limited, with no example of an HRT prescription model that incorporates the patient experience as an outcome and with no specific PREM available. This review also shows that European countries have been involved in HRT provision research from an early stage.

Most of the research on the assessment of HRT prescription models has been conducted within the last decade and mainly in European countries, highlighting the emergent interest and Europe’s leading position in this area of health research. In addition, HMV has attracted more attention from the scientific community in comparison with LTOT. Questionnaires were found to be the preferred method for data collection, however, existing databases from HRT registries or health services have also been used. Databases in comparison with questionnaires have the advantage of generating more representative data and may be a method of choice in future studies. The patient experience has not been examined in the assessment of the prescription models presented. While this reality was expected from the oldest studies, it was quite a surprising result for those from the last decade. These results show that, until now, the assessment of patients’ perceptions has not been seen as a priority in the assessment of prescription models. Unfortunately, this is also a reality in other health contexts and settings [10]. The Organisation for Economic Co-operation and Development (OECD) and Europe in “Health at a Glance: Europe 2018” reported critical gaps in the data on patient-reported experience, and they recommended collecting data on the patient experience from any doctor in ambulatory care settings [10]. Thus, future studies on the provision of HRT should address this important gap in the literature.

To address this gap, we need to be aware of the current methods being used to assess the patient experience. Different instruments used at distinct levels of healthcare are available and described in this review. These instruments were developed to be completed by adult patients and, in some cases, by carers of children. In our opinion, although the carers’ perspective is, of course, incredibly valuable, it should do not replace the children’s experience. The development of PREMs for pediatric populations is crucial to the collection of information on the experience and outcome of children’s care. Additionally, as previously mentioned, none of the instruments have been specifically designed to assess the patient’s experience with HRT. The development of a specific PREM for this health context should be a research priority in the upcoming years. The most commonly assessed domains in the described instruments, including the ERS/ELF survey, together with the key facilitators of the patient’s treatment experience, can be used as important sources of data to inform the development of a comprehensive instrument. Access to information and support, implementation of effective and clear communication, active participation in shared decision making, enhanced accessibility and navigability across the healthcare system for patients and families, particularly across transitional care, and management of polypharmacy are known to influence the patient experience in other healthcare settings and could be topics of interest to be included in future PREMs for patients on HRT [54]. Future studies should explore which of these raised topics are indeed meaningful for patients and carers.

On the basis of qualitative studies, it was found that education, training, support, and carer involvement were important key-points in facilitating the patient’s treatment experience and adherence. This knowledge comes mainly from the perspective of adult patients with COPD, pulmonary fibrosis, and OSA receiving CPAP and from their carers. These studies were conducted in five countries (three from Europe) [45,46,47,48,49,50,51,52,53]. Thus, this evidence may not completely apply to the experience of younger patients (including children) and that of their carers or to patients with other diseases and other treatment modalities (e.g., Bilevel Positive Pressure Airway, LTOT) and from other countries/continents. Considering these identified gaps, the experience of other patients receiving HRT could be explored in future studies. The identified key-points may inform the development process of semi-structured guides of focus groups or individual interviews to be used in these exploratory studies.

## 4. Conclusions

To the authors’ best knowledge, this is the first published work to review the emerging topic of the patient experience in the clinical context of HRT and give important insights into the status of this clinical research area while also pointing out possible directions in which to move to realize patient-centered care. The assessment of the patient experience is in its early stages, and further research is needed to integrate these measures with routine healthcare delivery and the core set of healthcare quality indicators, as well as and to drive quality improvements in HRT.

## Figures and Tables

**Table 1 jcm-08-00555-t001:** Studies assessing the prescription of home respiratory therapies.

Author, Year	Region or Country, Years Analyzed	Aim	Method	Data Collection	Results
Ekström et al., 2017 [20]	Sweden, 1987–2015	Long-term oxygen therapy (LTOT): incidence, prevalence, and the quality of prescription and management	Data from the Swedevox registry between 1 January 1987 and 31 December 2015	Data: Birth date, Sex, Primary/secondary causes of LTOT, Follow-up, Stop date and stop cause, PaO_2_ air and PaCO_2_ air, PaO_2_ oxygen and PaCO_2_ oxygen, FEV_1_ and VC, World Health Organization performance status, Height and weight, Never/Past/Current smoker, Maintenance treatment with oral corticosteroids, Oxygen dose, Oxygen duration.	23,909 patients on LTOT. 48 respiratory or medicine units. Incidence of LTOT increased from 3.9 to 14.7/100,000 inhabitants over the study time period. In 2015, 2596 patients had ongoing therapeutic LTOT in the registry, a prevalence of 31.6/100,000. Adherence to prescription recommendations and fulfilment of quality criteria were stable or improved over time. Of patients starting LTOT in 2015, 88% had severe hypoxemia and 97% had any degree of hypoxemia; 98% were prescribed oxygen for ≥15 hours/day; 76% had both stationary and mobile oxygen equipment; 75% had a mean PaO_2_ > 8.0 kPa breathing oxygen; and 98% were non-smokers.
Rose et al., 2015 [12]	Canada, 2012–2013	Home mechanical ventilation (HMV): national data profiling	Survey administered via a web link from August 2012 to April 2013 to service providers delivering care/services to ventilator-assisted individuals requiring daily noninvasive ventilation (NIV) or invasive mechanical ventilation via tracheostomy at home.	Survey content: provider characteristics, including services and education provided; user characteristics (age, ventilation type, primary disorder, duration of ventilation); criteria for initiation and monitoring ventilation effectiveness; equipment (ventilators and interfaces used, ventilator servicing arrangements and backup); training and education (audience, structure, topics, ongoing competency assessment); liaisons and transitions (referral, barriers to transition); follow-up (structure, frequency, location).	Response rate 152/171 (89%). 4334 ventilator-assisted individuals: an estimated prevalence of 12.9/100,000 population. 73% receiving NIV and 18% receiving intermittent mandatory ventilation (9% not reported). Services were delivered by 39 institutional providers and 113 community providers. Various models of ventilator servicing were reported. 64% of providers stated that caregiver competency was a prerequisite for home discharge, but repeated competency assessment and retraining were offered by 45%. Barriers to home transition: insufficient funding for paid caregivers, equipment, and supplies; a shortage of paid caregivers; negotiating public funding arrangements.
Escarrabill et al., 2015 [13]	Catalan Health Service (Spain), 2008–2011	HMV: prevalence and variability in prescriptions	Catalan Health Service (CatSalut) billing database, between 2008 and 2011.	Not reported (NR)	240,760 patients received some type of HRT funded by the public system. 75.8% used continuous positive airway pressure equipment, 17.3% used various forms of oxygen supply, 4.2% used nebulized therapy, 2.5% used HMV, and 0.2% used miscellaneous treatments. 6,867 patients received HMV, 23 users per 100,000 population. Rates of HMV increased by 39% over the study period
Nasiłowski et al., 2015 [14]	Poland, 2000–2010	HMV: trends over the last decade	Questionnaire designed specifically for the study was sent to the heads of nine HMV centers	Survey Content: Center details: location, area of activity (uniregional/multiregional), and year of initiating HMV. Number of subjects treated with HMV in each consecutive year. Overall number of treated subjects, divided into five disease categories: (1) neuromuscular diseases, (2) lung diseases (chronic obstructive pulmonary disease (COPD), bronchiectasis, cystic fibrosis, interstitial diseases), (3) chest-wall diseases (scoliosis, thoracoplasty, ankylosing spondylitis, post-tuberculosis sequelae), (4) hypoventilation syndromes (due to obesity, central congenital hypoventilation syndrome, central sleep apnea), (5) other diseases. Technique of ventilation (invasive and noninvasive). Number of new cases; Overall number of subjects treated with NIV or tracheostomy. Age of the treated subjects, Site where ventilation was initiated: intensive care unit, respiratory department, neurology department, general medicine department, home, or other.	Nine HMV centers, 1495 subjects Center experience 9 ± 3 years (6–13 years) One center was dedicated specifically to children, Two solely treated adults, and other centers treated subjects irrespective of age. In 2010, prevalence of HMV reached almost 2.5 subjects/100,000. The majority of subjects on HMV suffered from neuromuscular diseases (100% in 2000–2002 to 51% in 2010). Subjects with a diagnosis of respiratory failure due to pulmonary conditions appeared in 2004, and the number of subjects rapidly increased beginning in 2007. In 2010, they accounted for almost 25% of all HMV cases. Hypoventilation syndromes were the third main diagnostic group (4% until 2008, reaching 11% in 2010). Proportion of chest-wall diseases remained ~3%. In 2000 and 2001, ventilation via tracheostomy was exclusively used. The first subjects on NIV were treated in 2002. The number of subjects on NIV was 1/3 in 2004 and then leveled off for the following five years, followed by a rapid increase until 2010, when the proportions of subjects treated with NIV and tracheostomy equalized. Since 2008, the number of new cases treated noninvasively surpassed the number of new cases treated with invasive ventilation, and in 2010, the total number of subjects in both groups was virtually the same.
Garner et al., 2013 [15]	Australia and New Zealand, 2002–2004	HMV	HMV centers that had prescribed HMV for more than three months to more than five adult patients. A designed survey.	Survey Content: (1) Institutional details: location, type (e.g., tertiary), funding (e.g., government), patient catchment, years of service; (2) Criteria for HMV prescription by disease group (e.g., COPD); (3) HMV service details: number of patients receiving HMV, staffing levels, methods of implementation by location/tests utilized/staff involved, methods of follow-up by location/tests utilized/staff involved (0–3 grading from never to always), annual clinic attendances, presence of an outreach service; (4) Individual patient data (if available): age, gender, primary indication for HMV, duration of therapy, adherence to therapy, interface, machine settings (mode, inspiratory positive airway pressure, expiratory positive airway pressure, back-up rate); (5) Local database: current database for that center, data collected, what data should be collected, support for creation of a national database, center willing to participate; (6) Problems encountered with setting up an HMV service.	28 centers (82%) responded, providing data on 2725 patients. Prevalence of HMV was 9.9 patients/100,000 in Australia and 12.0 patients/100,000 in New Zealand. Variation existed among Australian states (range 4–13 patients/100,000) correlating with population density (*r* = 0.82, *p* < 0.05). The commonest indications for treatment were obesity hypoventilation syndrome (31%) and neuromuscular disease (30%). COPD was an uncommon indication (8%). No consensus on indications for commencing treatment was found.
Ringbaek et al., 2013 [21]	Denmark, 2001–2010	(LTOT: incidence, prevalence, treatment modalities, and survival in COPD.	Danish Oxygen Register in the period from 01 January 2001 to 31 December 2010: information on patients on home oxygen therapy, their prescriptions, and termination of therapy. National Health Services Central Register: information on diagnosis for LTOT and on vital status up to 31 December 2011.	NR	On 31 Dec 2001, a total of 2247 COPD patients (42.0/100,000) were receiving LTOT. The number of patients on LTOT had increased constantly to reach a prevalence of 48.1/100,000 in 2010. Incidence of oxygen therapy increased insignificantly from 30.5 to 32.2/100,000. The majority of COPD patients were women and older than 70 years of age. The mean age of patients who started LTOT during the study period increased from 73.4 ± 9 years to 74.8 ± 9.7 years. Most of the COPD patients were prescribed oxygen therapy by a hospital doctor immediately after an acute hospitalization, and the number of prescriptions from general practitioners was continuously declining toward zero during the study period. An increasing number of the COPD patients were prescribed oxygen at least 15 h daily and had delivered oxygen concentrator and mobile oxygen, whereas, in general, the oxygen flow remained low (≤1.5 L/minute). Compared with men, women started LTOT more often in connection with hospitalization and more often stopped LTOT within the first 6 months. Women were prescribed a lower oxygen flow than men and the treatment was more often specified to take place for 15–24 h per day.
Mandal et al., 2013 [16]	England, NA	HMV: prevalence of sleep and ventilation diagnostic and treatment services	A short survey delivered by email to 101 NHS Hospitals	Survey content: 10-item survey, focused on diagnostic services and HMV provision: (a) availability of diagnostics, (b) funding; (c) patient groups.	76 (68%) responses received; 42 (55%) trusts reported the provision of an HMV service. Only 65% of units charged for the delivery of an HMV service, with 12% of these services commissioned by an external provider. Median set-up frequency for the units charging was 42 patients per annum (interquartile range 23–73), whereas those units that failed to charge had a median of 11 (interquartile range 4–22). Of all the HMV set-ups, 67% were for obesity-related respiratory failure and COPD, with the other restrictive lung conditions forming the remainder
Serginson et al., 2009 [22]	Australia, 2004–2005	LTOT: prescription and costs	Data from all LTOT services in Australian Government’s departments and health services (state and federal) Centralized departments managing state budgets for LTOT provided costs (for the financial year 2004–2005) and patient numbers (point prevalence in 2005). If centralized data were not available, regional departments administering LTOT services were contacted.	Data: Costs were defined as “equipment only” (fees paid to oxygen companies) or “equipment and administrative” (wages and non-labor costs of administering programs included).	20,127 patients (100/100,000) through 59 different services at a cost of over $31 million. Prescription rates for LTOT per 100,000 population within each state ranged from 44 to 133, a threefold difference. Costs of LTOT per patient prescribed per year funded by individual states and territories ranged from $1014 to $2574. The cost of oxygen concentrators averaged $85 per month (range, $29–$109), portable oxygen ranged from $16 to $35 per month without refills, and, with a conserver included, $55 (two refills) to $166 unlimited refills) per month. All services provided concentrators for home use. Portable oxygen was funded in all states, except one (where it was limited to children and patients waiting for heart or lung transplants).
Jones et al., 2007 [23]	Tasmania (Australia), 2002–2004	LTOT	Records of all patients receiving Tasmanian Government-funded LTOT between December 2002 and April 2004	Data: Recipient demographics, Indications for LTOT, Oxygen prescription, Time to follow-up. The service provider provided usage reports and costs.	April 2004: 490 patients receiving LTOT Rate of 102/100,000; Median age at prescription of LTOT was 71.5 (range 0.7–97.2) years, and 54% of patients were female. Oxygen was prescribed for 267 patients (54%) during hospitalization, although only 192 of these patients (72%) met criteria for oxygen use at this time. LTOT was prescribed by respiratory physicians for 248 patients (51%) and by other hospital physicians for most of the remaining patients (39%). Data on indications were available for 430 patients (88%), and COPD accounted for 48% of prescriptions, but this proportion varied regionally. Median time to reassessment was 5.5 (range, 0.1–116) months, but varied between regions. Usage data were available for 175 patients (41%) using oxygen concentrators in April 2004. Of these 175 patients, 122 (70%) were prescribed oxygen for COPD. In this group, the median use was 18.3 (range, 0.38–24) hours per day; however, 36 (30%) had a median use < 15 hours/day.
Lloyd-Owen et al., 2005 [5]	16 European countries (Austria, Belgium, Denmark, Finland, France, Germany, Greece, Ireland, Italy, Netherlands, Norway, Poland, Portugal, Spain, Sweden, UK), 2001–2002	HMV: patterns of use across Europe	Questionnaire of center details, HMV user characteristics and equipment choices sent to selected HMV centers	Survey Content: Center (type of institution and year of starting HMV), Number of HMV users on 01 July 2001, Users’ characteristics (sex, age, and time on HMV). Users’ causes for respiratory failure: (1) Lung: lung and airway diseases: COPD, cystic fibrosis, bronchiectasis, pulmonary fibrosis, and pediatric diseases, including bronchopulmonary dysplasia; (2) Thor: thoracic cage abnormalities: early-onset kyphoscoliosis, tuberculosis sequelae such as thoracoplasty, obesity hypoventilation syndrome, and sequelae of lung resection; (3) Neur: neuromuscular diseases: muscular dystrophy, motor neuron disease (including amyotrophic lateral sclerosis), post-polio kyphoscoliosis, central hypoventilation, spinal cord damage, and phrenic nerve paralysis. Type of ventilator and interface used.	329 centers completed surveys, 21,526 HMV users; Estimated prevalence of HMV was 6.6/100,000 in the 16 European countries. Differences between countries in the relative proportions of (1) lung and neuromuscular patients using HMV and (2) the use of tracheostomies in lung and neuromuscular HMV users. Lung users were linked to an HMV duration of <1 year, thoracic cage users with 6–10 years of ventilation and neuromuscular users with a duration of ≥6 years. Almost all of the HMV users had positive pressure ventilators, with only 0.005% (79 users) having other types. Volume preset positive pressure ventilators were used the least for lung problems and most frequently for neurological problems (% volume: Lung 15%; Thor 28%; Neur 41%). Overall, 13% of the survey population had ventilation via a tracheostomy with the highest percentage in neuromuscular patients (Neur 24%; Thor 5%; Lung 8%).
Chu et al., 2004 [17]	Hong Kong (China), 2002	HMV	Survey to consultants of respiratory medicine in all adult medical departments of Hong Kong Hospital Authority hospitals to report their adult patients (>18 years) who had ever been managed by HMV	Survey content: demographic data, mode of ventilation (non-invasive or tracheostomy ventilation), underlying disease, indications for HMV, time of starting ventilation, time and reason of stopping ventilation, if any, in the follow-up period.	249 cases reported to the survey from 14 centers of adult respiratory medicine; 156 males (62.7%) and 93 females (37.3%) with a mean age of 62.7 ± 13.8 years; 80% of HMV cases were under the care of six major centers. 197 cases were continuing with HMV, corresponding to ~2.9 HMV users per 100,000 population. The majority (*n* = 236, 94.8%) were treated by noninvasive ventilation (NIV), with the remaining 13 patients (5.2%) receiving tracheostomy ventilation. All NIVs were provided by bilevel pressure-support ventilators. All tracheostomized cases were put on HMV after repeated failures to wean. The disease conditions for which HMV was prescribed: COPD (121, 48.6%); Complicated obstructive sleep apnea/obesity hypoventilation syndrome (43, 17.2%); and Restrictive thoracic disorders (85, 34.1%).
Fauroux et al., 2003 [18]	France, 2000	Domiciliary non-invasive mechanical ventilation (NIMV) in children	Anonymous national cross-sectional Survey A postal questionnaire sent by the Paediatric Group of the National Home Care Organization (ANTADIR) in 1999 to all 64 senior pediatric respiratory, neurology, and intensive care physicians in France. Patients aged < 18 years and receiving home NIMV were included in the study.	All physicians taking care of children with NIMV were sent a second questionnaire in 2000. The specific information requested on each patient included: Sex and date of birth; Primary and secondary diagnosis; Symptoms that justified NIMV; Age at onset of NIMV; Type of nasal mask, ventilatory mode, and concurrent use of oxygen therapy; Investigations performed before initiating of NIMV and during follow-up.	102 patients from 15 centers: 4/15 centers cared for 84% of patients; 7% of patients were under 3 years; 35% were 4–11 years; and 58% were >12 years. Underlying diagnoses included neuromuscular disease (34%), obstructive sleep apnea and/or craniofacial abnormalities (30%), cystic fibrosis (17%), congenital hypoventilation (9%), scoliosis (8%), and other disorders (2%). NIMV was started because of nocturnal hypoventilation (67%), acute exacerbation (28%), and/or failure to thrive (21%). Volume-targeted ventilation was preferred in restrictive disorders (56%) and central hypoventilation (56%), while pressure support ventilation (PSV) was preferred in cystic fibrosis (71%). Patients with obstructive sleep apnea and/or craniofacial abnormalities were ventilated with continuous positive airway pressure (45%) or bilevel PSV (52%).
Wijkstra et al., 2001 [24]	Seven countries (Brazil, Canada, France, Italy, Spain, Netherlands, USA), NR	LTOT: prescription	Questionnaire mailed to 100 randomly selected respirologists from a list of respiratory specialists belonging to a professional organization in each country	Characteristics of the respirologists: Date of birth; How many years they had been practicing respiratory medicine; Number of patients for whom they prescribed oxygen for the first time or for renewal purposes over the previous month. Prescription of oxygen at rest; Whether they prescribed a standard oxygen flow rate for all their patients or whether they individualized flow rates with or without specific testing of each patient; How the recommended oxygen flow at rest was chosen (either tested at rest or tested during exercise); The position (sitting, semirecumbent, supine) in which the patients were tested, the target level of arterial oxygen saturation (SaO_2_) used to establish an oxygen prescription and the percentage of time during the measure in which this target had to be achieved. Prescription of oxygen during sleep and exercise; How they prescribed oxygen during sleep and exercise; The type of exercise test (walking, laboratory testing) used to establish the exercise prescription; The target level of saturation during exercise and the percentage of time during the test in which this target had to be achieved.	81% of respondents individualized the oxygen prescription at rest. Resting SaO_2_ was most commonly targeted at 90–91%. The approach to night prescription varied. Respirologists in Canada and the USA increased the resting SaO_2_ by 1–2 L/min during sleep, while those in Spain used the resting flow for the night prescription (62%). Respirologists in the Netherlands, France, and Italy individualized the night prescription more frequently. Although oxygen during exercise was individualized in most countries (74%), significant differences remained among countries. 62% of respirologists (62%) aimed to achieve an SaO_2_ of 90–91% during exercise, while 70% of all respirologists tried to achieve the desired SaO_2_ for 90% of the test.
de Lucas Ramos et al., 2000 [4]	Spain, 1998–1999	HMV: prescription	Questionnaire mailed to the respiratory medicine departments of 200 hospitals in the public health system	Survey Content: Center name, Year of initiation of the MV program, Number of patients in the first year, Number of patients in the current year. Diagnosis: Neuromuscular disease, Thoracic cage disease, Hypoventilation-obesity syndrome, COPD, Other. Ventilation type: Volumetric, BI-level Positive Airway Pressure, Interface, Nasal mask, Conventional, Personalized, Tracheostomy, Mouthpiece.	43 hospitals, 1821 patients; 813 patients had restrictive disease due to thoracic cage disease, 452 neuromuscular disease, 271 hypoventilation-obesity syndrome, 162 COPD, and 123 other diseases/conditions. 965 (53%) used pressure support devices and 856 (47%) used volumetric ventilators. 1320 conventional nasal mask, 336 personalized nasal mask, 118 tracheostomy, 41 facial mask, six mouthpiece.
Fauroux et al., 1994 [19]	13 European countries (Belgium, Denmark, England, France, Germany, Ireland, Italy, Netherlands, Norway, Poland, Spain, Sweden, Switzerland), 1992	Home care of chronic respiratory insufficiency	Questionnaire at the end of 1992.	Questionnaire content: Home treatments (LTOT, HMV); Prescribers; Practical organization of home care (supply of material, supervision of patients and equipment). Information on patients: diagnostic information (either obstructive, restrictive, or mixed pulmonary disease); Age; Sex; Equipment supplied; Service provided; Therapeutic schedules.	Information was easier to obtain for LTOT than for HMV. In all countries, both adults and children received LTOT at home for lung diseases and other less common problems, such as chest-wall deformities and sequelae of tuberculosis. Oxygen concentrators were used preferentially in all countries except Italy (80% of the patients received liquid oxygen), Denmark, Spain, and the Netherlands (cylinders were used by 80% of the patients). Both adults and children received HMV at home for chronic lung disease, neuromuscular disease, chest-wall deformities, and central hypoventilation in all countries, except in Denmark and Poland, where this treatment is almost unknown in the home. Home ventilator treatment was generally performed by volume-cycled ventilators. National prescription rules existed in some parts of Spain, Switzerland, and Belgium. In other countries, such as Germany, prescriptions relied on recommendations elaborated by specialists or international guidelines. Service and equipment were provided by national organizations, health services, commercial companies, or hospitals. Home supervision of the patient was performed by a nurse and/or a doctor and equipment maintenance by a technician.

**Table 2 jcm-08-00555-t002:** Instruments designed to assess patient’s experience with the provision of care.

Instrument	Population	Setting	Language	Concepts	Structure	Measurement Properties
CEFIT: Care Experience Feedback Improvement Tool [36]	Tested in 802 patients (≥18 years) with healthcare experience.	Hospital	English	Safe, Timely, System navigation, Caring, Effective.	Five questions scored using a five-point scale, from 1 (never) to 5 (always).	Reliability Validity
COPD PREM9: disease-specific patient-reported experience measure in COPD [28,30]	Tested in 174 adult patients with COPD.	Clinical settings (e.g., pulmonary rehabilitation, nurse-led clinics, or GP annual reviews)	English	Everyday life with COPD, Everyday care in COPD, Self-management of COPD, exacerbations.	Nine questions scored using a six-point scale, from 0 (good experience) to 5 (bad experience).	Reliability Validity
GS-PEQ: Generic Short Patient Experiences Questionnaire [31]	Tested in 1324 patients (including outpatients undergoing rehabilitation and carers of children).	Services provided in a range of specialist healthcare (in- and out-patient)	Norwegian	Outcome, Clinician services, User involvement, Incorrect treatment, Information, Organization, Accessibility.	10 questions scored using a five-point scale, from 1 (Not at all/Not important) to 5 (To a very large extent/Of utmost importance).	Not reported
howRwe (how are we doing?)questionnaire: short generic patient experience questionnaire [32,43]	Tested in 828 patients in an orthopedic pre-operative assessment clinic [32] and in 90 adult patients (≥18 years) from general practices (10 with COPD) [43].	Generic, applicable without change across all patient categories and care settings, including primary, secondary, community, emergency, domiciliary, and social care.	English Dutch	Clinical care (kindness and communication), Organization of care (promptness and organization).	Four items scored using a four-point scale from 0 (poor) to 3 (excellent).	Reliability Validity
Health Services OutPatient Experience (HSOPE): global outcome measure of perceived patient-centeredness of the outpatient healthcare pathway [35]	Tested in 1532 adult outpatients (≥16 years) receiving care (including rehabilitation).	Hospital	Italian	Perceived technical effectiveness of the staff, Information on modalities of the outpatient visit, on the visit outcomes, and the course of the healthcare pathway, Relational aspects of outpatient–staff interaction, Involvement in decision making.	10 statements scored using a five-point Likert scale from 1 (never) to 5 (always) 1 item scored using a 10-point scale from 1 (very dissatisfied) to 10 (very satisfied). Three sociodemographic questions (sex, age, and residence). One question about suggestions to improve outpatient visits.	Reliability Validity
Intermediate care-IC-PREMs: Bed-Based Patient-Reported Experience Measure [41]	Tested in 1832 adult patients.	Bed-based IC services	English Italian	Goal Setting, Empowerment, Self-Management, Care-Planning, Transitions, Decision Making, Communication.	15 questions scored using two, three, or four response categories.	Reliability Validity
IC-PREMs: home-based (and reablement-based) Patient-Reported Experience Measure [41]	Tested in 4627 adult patients.	Home-based or reablement IC services	English Italian	Goal Setting, Empowerment, Self-Management, Care-Planning, Transitions, Decision Making, Communication.	15 questions scored using two, three, or four response categories.	Reliability Validity
IEXPAC, Instrument for Evaluation of the Experience of Chronic Patients [29]	Tested in 356 patients (≥16 years) with chronic diseases (20% with COPD).	Health and social services	Spanish	Type and scope of patient and professional interactions oriented to patient activation. Patient’s self-management capacity of his/her wellbeing resulting from the interventions received. New relational model of the patient with the system through the internet or with partners in group intervention.	11 + 1 items scored using a five-point scale from 0 (never) to 10 (always). Since 2018, a new version with 11 + 4 items is used, with three additional items.	Reliability Validity
LifeCourse experience tool [33]	Tested in 607 adult patients with emergency department and in-patient utilization, advanced primary diagnosis of heart failure, cancer, or dementia.	Home, Nursing Homes, Assisted living	English	Care Team, Communication, Care Goals.	22 items scored using a four-point scale from 1 (Never or Strongly Disagree) to 4 (Always or Strongly Agree).	Reliability Validity
Multidimensional Semantic Patient Experience Measurement Questionnaire [37]	Tested in 60 patients (≥15 years) undergoing a magnetic resonance scan.	Hospital	English	Evaluation/valence, Potency/control, Activity/arousal, Novelty.	12 rating scales using a seven-point bipolar attribute rating scales: ‘extremely’, ‘quite’, ‘slightly’, ‘neither’, ‘slightly’, ‘quite’, and ‘extremely’.	Reliability
PEQ: Patient experience questionnaire 2001 [39]	Tested in 1092 patients (1–91 years)/carers	Primary care	Norwegian	Communication, Emotions, Short-term outcome, Barriers, Relations with auxiliary staff.	Total 18 items: Four items using a five-point scale from 1 (‘no more’ or ‘nothing’) to 5 (‘much more’ or ‘a lot’). 10 items using a five-point scale from 1 (disagree completely) to 5 (agree completely). Four items were formed on seven-point scales.	Reliability Validity
PEQ: Patient Experiences Questionnaire 2004 [38]	Tested in 19578 patients (≥16 years) with experience with surgical wards and wards of internal medicine	Hospital	Norwegian	Information on future complaints, Nursing services, Communication, Information examinations, Contact with next-of-kin, Doctor services, Hospital and equipment, Information medication, Organization, General satisfaction.	35 items with 10-point ordinal response scales from 1 (negative) to 10 (positive).	Reliability Validity
PACIC: Patient Assessment of Chronic Illness Care [34,44]	Tested in 4108 adult patients with diabetes, chronic pain, heart failure, asthma, coronary artery disease.	Chronic care management	English	Patient activation, Delivery system design, Goal setting, Problem solving, Follow-up/coordination. Focuses on the receipt of patient-centered care and self-management behaviors.	20 items using a five-point scale from 1 (Almost Never) to 5 (Almost Always).	Reliability Validity
ACES-SF: Ambulatory Care Experiences Survey [40]	Tested in 49,861 adult patients.	Primary care	English	Quality of physician–patient interaction, Health promotion support, Care coordination, Organizational access, Office staff interactions, An additional item to assess patients’ willingness to recommend the physician to family and friends.	18 items using continuous responses: Never, Almost never, sometimes, Usually, Almost always, Always; or Yes, definitely, Yes, somewhat, No, definitely not; or Definitely yes, Probably yes, Not sure, Probably not, Definitely not.	Reliability Validity

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
