# Peer review of "Patient Experience in Home Respiratory Therapies: Where We Are and Where to Go"

_jcm, 2019, doi:10.3390/jcm8040555_

Reviewer 1 Report

PEER REVIEW ON: Patient Experience in Home Respiratory Therapies: where we are and where to go

JCM-468145

This is a narrative review describing patient experiences with home respiratory therapies such as home oxygen therapy and home mechanical ventilation. It is mainly from a European perspective (with strong Portugal influence), but other perspectives were included. It starts by describing tools to assess patient experience with questionnaires and by a qualitative description of studies using focus groups or interviews. It ends with summarizing studies investigating epidemiology of home oxygen and HMV i.e. prevalence and incidence.

I was happy to see a review like this as the patient experience is definitely important, especially for community-based care. It was great to see a description of some of the instruments and themes that clinicians and researchers may focus on to help address this gap.

Major comments

·        Section 3 describing prescription of models of HRT is misplaced in the review. The objective of the review was to describe the patient experience, and the authors even state (line 168-169) “none of the 15 studies reported patient experience with HRT”. Similarly, in the discussion authors state there was no examples of an HRT prescription model incorporating patient experience as an outcome. The objective of 15 studies presented was not to describe the patient experience but to describe the burden or landscape of home oxygen and HMV - this is why it is “lacking” in the studies. The question of patient experience would come from other studies – the ones assessing instruments and utilizing qualitative methodologies (Section 2). Suggest focussing more on and expanding on the sections describing instruments and qualitative studies, and remove Section 3. Or briefly describe the contents of Section 3 in the background or beginning of the paper i.e. describe the “burden” of HRT (15 prevalence/incidence studies) à patient experience instruments + Table 1 à qualitative studies + create Table à Discussion on how to address gaps in patient experience for HRT.

·        To expand on previous point - under Section 2.1.2 expand on how carers help patients manage their disease? The carer role is likely one of the largest contributions to the patient experience since they interact on a day-to-day basis. The description of carers could be expanded to relate the importance.

·        Expand and focus the discussion on information that have definitely addressed HRT patient experience. And describe MORE on where to focus for future research. Much of the descriptions were only touched on. For example, line 186 to 189 “Future works should also address this issue. Information, communication, shared decisions, accessibility, navigability across healthcare system, transitional care and management of polypharmacy could be areas of interest to define PREM's in HRT [53].” Each of these items should be described in more detail. This will help future research.

·        Not sure why the focus is on Portugal, when many of the studies take place in other countries, and the information would be relevant elsewhere. Many readers will likely not be interested in knowing if it’s relevant in Portugal or not. Making the review applicable to Portugal can still be accomplished by describing the review generically.

Minor comments:

·        Descriptions on a search strategy would be helpful i.e. how did they find the studies to put into the review? And how did they decide on which studies/instruments to describe in the review?

·        Suggest separating home oxygen therapy and home mechanical ventilation in the descriptions.

·        Please specify the population of focus – adults and children? If both (though would recommend focussing on one or the other), describe which studies looked at adults and which included children.

·        Section 2.1.1 and Table 1 – I am assuming all those instruments were tested for validity and reliability? Could you put a line in that section stating this?

·        Avoid using “recent” when describing studies. Some studies are getting close to 10 years old.

·        This review was generally written quite well, but there were a few minor issues that I am assuming will be cleaned up during the editing process. For example, line 119 “In the perspective of patients” would read better as “Based on the perspectives of patients…”. Line 121 (e.g., COPD, OSA,…) could be (e.g. COPD, OSA). Line 146 to 149 is quite a long sentence.

Author Response

Reviewer 1

We thank the reviewer for the time spent considering our paper and for his/her comments. We have made an attempt to address his/her recommendations. Below we present the list of amendments that have been performed, following the list of comments.

Major comments:

Section 3 describing prescription of models of HRT is misplaced in the review. The objective of the review was to describe the patient experience, and the authors even state (line 168-169) “none of the 15 studies reported patient experience with HRT”. Similarly, in the discussion authors state there

was no examples of an HRT prescription model incorporating patient experience as an outcome. The objective of 15 studies presented was not to describe the patient experience but to describe the burden or landscape of home oxygen and HMV - this is why it is “lacking” in the studies. The

question of patient experience would come from other studies – the ones assessing instruments and utilizing qualitative methodologies (Section 2). Suggest focussing more on and expanding on the sections describing instruments and qualitative studies, and remove Section 3. Or briefly

describe the contents of Section 3 in the background or beginning of the paper i.e. describe the “burden” of HRT (15 prevalence/incidence studies) à patient experience instruments + Table 1 à qualitative studies + create Table à Discussion on how to address gaps in patient experience for HRT.

Thank you for your pertinent suggestion. We agree that it is more interesting to start the review by presenting the current status/burden of HRT across distinct countries (now section 2.1), working as an introduction of the main topics. Only then describe the methodologies available to assess patient experience (now section 2.2), which is the main focus of this work.

To expand on previous point - under Section 2.1.2 expand on how carers help patients manage their disease? The carer role is likely one of the largest contributions to the patient experience since they interact on a day-to-day basis. The description of carers could be expanded to relate the importance.

A more detailed description of the carer role and how they can help patients managing their disease and their HRT equipment has now been added. Please see page 18, lines 186-195.

Expand and focus the discussion on information that have definitely addressed HRT patient experience. And describe MORE on where to focus for future research. Much of the descriptions were only touched on. For example, line 186 to 189 “Future works should also address this issue.

Information, communication, shared decisions, accessibility, navigability across healthcare system, transitional care and management of polypharmacy could be areas of interest to define PREM's in HRT [53].” Each of these items should be described in more detail. This will help future research.

Thank you for your observation. The discussion of these topics has now been expanded and recommendations for future research made. Please see page 19, lines 229-235.

Not sure why the focus is on Portugal, when many of the studies take place in other countries, and the information would be relevant elsewhere. Many readers will likely not be interested in knowing if it’s relevant in Portugal or not. Making the review applicable to Portugal can still be

accomplished by describing the review generically.

Thank you for your valuable comment. We understand your point of view and we totally agree. The specific references to Portugal data/participation have now been removed.

Minor comments:

Descriptions on a search strategy would be helpful i.e. how did they find the studies to put into the review? And how did they decide on which studies/instruments to describe in the review?

Thank you for your comment. A short description of the methodology employed has now been added. Please see page 2, lines 69-80.

Suggest separating home oxygen therapy and home mechanical ventilation in the descriptions.

Thank you for your comment. We included a short reference in section 2.1 (page 2, line 84) and 2.2.2 (page 18, lines 150-152) reporting which studies focussed on HMV and on LTOT. We also have now discussed the fact that more research is available for HMV in comparison with LTOT both regarding the prescription models (page 19, lines 204-205) and patient experience (page 19, lines 241-242).

Please specify the population of focus – adults and children? If both (though would recommend focussing on one or the other), describe which studies looked at adults and which included children.

In the beginning of our search, we aimed to address both paediatric and adult populations, but as we were writing we understood that the majority of evidence was from adults. In each section, we have now described if evidence is available for the two populations. Please see pages 2 (lines 93-94), 13 (lines 124-126) and 18 (line 154).We also added a discussion of this limitation. Please see page 19, lines 240-241.

Section 2.1.1 and Table 1 – I am assuming all those instruments were tested for validity and reliability? Could you put a line in that section stating this?

You are right. All these instruments were published and had some of their psychometric properties studied. A column was added to Table 2 (former table 1) stating what psychometric properties were analysed for each instrument. Please see table 2.

Avoid using “recent” when describing studies. Some studies are getting close to 10 years old.

Thank you for your pertinent comment. The expression “recent” has been removed.

Reviewer 2 Report

Thank you for the opportunity to contribute to the peer review process for the original submission manuscript entitled ‘Patient Experience in Home Respiratory Therapies:  where we are and where to go’ (jcm-468145) from Caneiras et al. The authors have presented the outcomes of their review of assessment of patient experience in the clinical context of Home Respiratory Therapies, and in particular highlighting the European and specifically the Portuguese contribution for this emerging body of knowledge. The authors conclude that development of patient-reported experience measures, and the integration of such into patient assessments and prescription models should be a priority.

However, the one major limitation is that for a review article the authors do not describe their methodology. Therefore, the absence of specific details as to how and why the articles discussed came to be included opens the door to potential concerns of inclusion and/or reporting bias. It is recommended that the authors include some commentary as to the process / strategy they used to conduct their review. Whilst accepting this is may not be intended as a systematic review, some description of approach undertaken is indicated.

Otherwise the manuscript is well structured and written, with relatively minor suggested editorial type amendments only offered, as below.

Specific comments

Abstract, line 34 – suggest amending tense to ‘…Development of a … and integration of PREMs…’

Introduction, line 45– suggest amending to avoid starting sentence with an abbreviation ‘Thus, HRT is posing…’

Line 56 – suggest rephrasing to ‘However, these metrics...’

Line 62 – suggest adding a comma ‘…suggests, the patient…’

2.1.2, line 112 – suggest utilizing previously defined abbreviation here ‘…and OSA, specifically...’

Line 118 – suggest rephrasing to depersonalize ‘Below is described in…’

Line 139 – suggest adding a space ‘adherence [39]’

Line 141 – suggest adding a space ‘therapy [34]’

Line 142 – suggest adding a space ‘weeks [32,36]’

Line 144 – suggest adding a space ‘hotline [34]’

Discussion, line 176 – suggest supplementing to ‘…as an outcome.’

Line 201 – suggest amending to ‘…HMV [5], …’

Conclusion, line 215 – suggest amending to ‘…first published work…’

References

There are numerous instances in the reference list where capitalization review is required – for example countries / continents / cities (line 241 Europe), entities (line 248 NHS), and abbreviations (line 249 OECD, EU).

Further there is inconsistency in journal title abbreviation and capitalization – recommend review and harmonization with style guide.

Line 353, this citation is missing details of source - ?journal title, pages etc

Author Response

Reviewer 2

We thank the reviewer for the time spent considering our paper and for his/her comments. We have made an attempt to address his/her recommendations. Below we present the list of amendments that have been performed, following the list of comments.

Comments

However, the one major limitation is that for a review article the authors do not describe their methodology. Therefore, the absence of specific details as to how and why the articles discussed came to be included opens the door to potential concerns of inclusion and/or reporting bias. It is recommended that the authors include some commentary as to the process / strategy they used to conduct their review. Whilst accepting this is may not be intended as a systematic review, some description of approach undertaken is indicated.

Thank you for your valuable comment. We have now added a paragraph explaining the methodology employed. Please see page 2, lines 69-80.

Specific comments

Abstract, line 34 – suggest amending tense to ‘…Development of a … and

integration of PREMs…’

Thank you for your suggestion, which we have followed. Please see page 1, line 33.

Introduction, line 45– suggest amending to avoid starting sentence with an abbreviation ‘Thus, HRT is posing…’

The sentence has now been modified accordingly. Please see page 1, line 45.

Line 56 – suggest rephrasing to ‘However, these metrics...’

The sentence has been rephased accordingly. Please see page 2, line 56.

Line 62 – suggest adding a comma ‘…suggests, the patient…’

A comma has now been added. Please see page 2, line 62.

2.1.2, line 112 – suggest utilizing previously defined abbreviation here ‘…and OSA, specifically...’

We have followed your suggestion. Please see page 18, line 149.

Line 118 – suggest rephrasing to depersonalize ‘Below is described in…’

We have followed your suggestion. Please see page 18, line 158.

Line 139 – suggest adding a space ‘adherence [39]’

We have followed your suggestion. Please see page 18, line 180.

Line 141 – suggest adding a space ‘therapy [34]’

We have followed your suggestion. Please see page 18, line 182.

Line 142 – suggest adding a space ‘weeks [32,36]’

We have followed your suggestion. Please see page 18, line 183.

Line 144 – suggest adding a space ‘hotline [34]’

We have followed your suggestion. Please see page 18, line 185.

Discussion, line 176 – suggest supplementing to ‘…as an outcome.’

We have followed your suggestion. Please see page 19, line 199.

Line 201 – suggest amending to ‘…HMV [5], …’

This sentence has now been deleted.

Conclusion, line 215 – suggest amending to ‘…first published work…’

Thank you for your suggestion, which has been inserted. Please see page 20, line 248.

References

There are numerous instances in the reference list where capitalization review is required – for example countries / continents / cities (line 241 Europe), entities (line 248 NHS), and abbreviations (line 249 OECD, EU).

Thank you for your observation. References have now been reviewed and capitalization standardized. Please see references list (pages 20-23).

Further there is inconsistency in journal title abbreviation and capitalization –

recommend review and harmonization with style guide.

Journal title abbreviations have now been standardized. Please see references list (pages 20-23).

Line 353, this citation is missing details of source - ?journal title, pages etc

This reference is now complete. Please see reference 54.

Round  2

Reviewer 2 Report

Thank you for the opportunity to contribute again to the peer review process for the revised original submission manuscript entitled ‘Patient Experience in Home Respiratory Therapies:  where we are and where to go’ (jcm-468145_V2) from Caneiras et al. The authors have presented the outcomes of their review of assessment of patient experience in the clinical context of Home Respiratory Therapies, and in particular highlighting the European and specifically the Portuguese contribution for this emerging body of knowledge. The authors conclude that development of patient-reported experience measures, and the integration of such into patient assessments and prescription models should be a priority.

The authors have outlined their response to the review process, new changes are evident in the revised manuscript in response to the both reviewer's recommendations, and the manuscript has evolved positively via this review process.

The following minor recommendations are suggested to be incorporated as part of the acceptance/proofing process:

lines 70-80 - somewhere in this paragraph should add a comment as to whether there was any date limiter to the search strategy - where only articles from last XX years considered, for example?

line 116 - review phrase '...are abler and...' as this does not quite make sense / read well for this reviewer.

Author Response

We thank the reviewer for the time spent considering our paper and for his/her comments. We have made an attempt to address his/her recommendations. Below we present the list of amendments that have been performed, following the list of comments.

lines 70-80 - somewhere in this paragraph should add a comment as to whether there was any date limiter to the search strategy - where only articles from last XX years considered, for example?

Thank you for your comment. This information has now been added to the manuscript. Please see page 2, lines 79-80.

There was no time restriction in the literature search, although it was limited to English, Portuguese, or Spanish.”

line 116 - review phrase '...are abler and...' as this does not quite make sense / read well for this reviewer.

This sentence has now been rewritten. Please see page 13, line 116.

but PREMs are more adequate to assess experience with healthcare.